# An analysis on the effects of speaker embedding choice in non auto-regressive TTS

*Adriana STAN* [1], *Johannah O'MAHONY* [2]

[1]Communications Department, Technical University of Cluj-Napoca, Romania
[2]Centre for Speech Technology Research, University of Edinburgh, United Kingdom
`adriana.stan@com.utcluj.ro, johannah.o'mahony@ed.ac.uk`

## Abstract

In this paper we introduce a first attempt on understanding how a non-autoregressive factorised multi-speaker speech synthesis architecture exploits the information present in different speaker embedding sets. We analyse if jointly learning the representations, and initialising them from pretrained models determine any quality improvements for target speaker identities. In a separate analysis, we investigate how the different sets of embeddings impact the network's core speech abstraction (i.e. zero conditioned) in terms of speaker identity and representation learning. We show that, regardless of the used set of embeddings and learning strategy, the network can handle various speaker identities equally well, with barely noticeable variations in speech output quality, and that speaker leakage within the core structure of the synthesis system is inevitable in the standard training procedures adopted thus far.

**Index Terms**: speech synthesis, speaker embeddings, multi-speaker TTS, speaker disentanglement, speaker verification, non-autoregressive TTS, factorised TTS

## 1. Introduction

Multi-speaker Text-to-Speech (TTS) has gained significant attention in recent years due to its potential applications in various fields such as entertainment, education, and accessibility. One of the key challenges in multi-speaker TTS is to accurately model the acoustic characteristics of different speaker identities (e.g. timbre, prosodic and phonetic particularities). This is usually performed by conditioning the deep neural network on a low-dimensional representation. This representation can either be *learned jointly* with the TTS architecture [1, 2, 3], or transferred from an *external neural network* [4, 5]. These representations are commonly stored as a lookup table (an embedding layer) which can be interrogated at inference time to generate the desired speaker identity.

When jointly learning speaker embeddings, the speakers in a multi-speaker TTS corpus are mapped to an index which is embedded, with the resulting embedding used to condition the utterances of that target speaker during training. As a consequence of this, the learned embeddings are restricted to the speaker identities in the TTS training corpus. Using jointly-learned embeddings, however, isn't suitable for all training objectives. For example, when building architectures which are able to perform zero- or few-shot adaptations towards an unseen target speaker, we are more heavily reliant on the choice of embedding, as using jointly learned embeddings does not enable the network to add novel representations for unseen speaker identities.

To account for this, external speaker representations can be used which have been trained on more speakers than seen during TTS model training. The most common external representations used in multi-speaker TTS models are derived from speaker verification (SV) networks [6, 7, 8]. In SV networks the objective is to maximise speaker discrimination over a large pool of speaker samples. Though it is assumed that the output representations can separate the speaker identity from other factors in the speech signal, such as language, background noise, and speaking style, this assumption is not entirely accurate [9]. This results from most speakers in the training set having only a few samples recorded in a single environment and at a specific time. Therefore, any characteristics unique to the speaker's audio in the training set are likely to be contained in the resulting representation, not just the speaker identity in itself.

Even if the SV networks truly disentangled speaker representations, the manner in which these representations are incorporated into the TTS models may reduce their overall effect and contribution to the output speech. In this paper we investigate this effect using FastPitch, a non-autoregressive factorised text-to-mel synthesis model [10]. Six different sets of speaker representations and their learning processes are examined with respect to the changes they produce in the output speech, as well as to how they influence the abstraction of the speech within the core network. A separate set of analyses looks into potential means of removing the speaker influence in the core network by freezing modules in the architecture.

## 2. Related work

There are several papers which have already examined the use of external speaker embeddings in conditioning the output of a TTS model, for either multi-speaker systems, or for zero- or few-shot adaptation scenarios. [6] was one of the first studies to propose the use of speaker embeddings separately learned and then transferred to the TTS model. [4] explores the use of i-vectors [11], x-vectors [12] and learnable dictionary encodings in a Tacotron2 TTS system for zero-shot speaker adaptation.

[13] builds upon the VITS model and adds several novel modifications for zero-shot multi-speaker and multilingual training. It uses external speaker embeddings based on the Clova AI H/ASP model [14]. [15] explores low-resource and zero-shot settings for multi-speaker TTS in a FastSpeech architecture. The embeddings are extracted with a ECAPA-TDNN [7] or based on x-vectors.

In some of the most recent models, large multi-speaker datasets are used for pretraining the network, and as a result, the speaker embedding is also jointly learnt, yet it is able to accommodate a larger number of speakers [16].

To the best of our knowledge, there are no studies which explore how, and if, the choice of speaker representations and their learning strategies affect the synthesised output in multi-speaker

TTS. One study which performed a related task on understanding what the SV-derived embeddings encompass is [17]. The study introduced an evaluation over six freely available neural speaker embedding architectures and the extent to which they encompass residual information related to other speech factors, such as $F_0$, duration, signal-to-noise ratio, speaker gender, and linguistic content. Its results showed that there is still a large amount of information which is not a direct representation of a speaker's spectral and prosodic particularities, but rather a wider representation of signal characteristics of the speaker samples available at training time.

# 3. Experimental setup

## 3.1. Speech data

In order to test the behaviour of the network for various speaker identities, and to factor out, to some extent, the linguistic content, we use a parallel set of samples from the Romanian SWARA corpus [18]. 18 speakers were selected, 8 male and 10 female, and each of them read aloud the same set of 712 prompts. All speakers were recorded in studio conditions at 48kHz. The data was downsampled to 22kHz at 16 bits per sample. The total duration of the dataset is 12 hours and 32 minutes. Depending on the speech rhythm, the duration for each speaker's data is between 35 and 56 minutes. 200 samples from this parallel set of prompts were set aside for evaluation purposes.

## 3.2. Pretrained models

To replicate the common approach to multi-speaker TTS, we first pretrained a model using the SWARA2.0 corpus which includes 28 speakers each reading aloud between 1600 and 1800 utterances in their home environments. There are a total of 50,042 samples amounting to approximately 60 hours of speech data. We adopt a TTS architecture which enables the individual control of various speech factors, and is non-autoregressive. Our multispeaker TTS system uses the Fast-Pitch architecture [10] and its official implementation.[1] Fast-Pitch is a transformer-based network with distinct modules for pitch, duration and energy prediction. In multi-speaker training scenarios, a speaker representation is summed to the text embedding and fed to the encoder. The encoder's output conditions all the variance adaptors for pitch, duration and energy. The summation of all these encodings is then passed through the decoder layers.

For the pretrained model we did not use speaker conditioning. The model was trained for 6M iterations with a batch size of 16. A separate model trained on a single female speaker was also included in the analysis. The model uses the complete Mara Corpus [19] which contains over 11 hours of professionally recorded speech of a Romanian novel. The model was trained for 1M iterations with a batch size of 16.

The output waveforms were generated using a pretrained HiFi-GAN [20] vocoder.[2] No finetuning of the released model was performed.

[1]https://github.com/NVIDIA/
DeepLearningExamples/tree/master/PyTorch/
SpeechSynthesis/FastPitch
[2]https://github.com/NVIDIA/
DeepLearningExamples/tree/master/PyTorch/
SpeechSynthesis/HiFiGAN

## 3.3. Speaker embedding sets and multi-speaker models

Six different embedding sets and embedding learning strategies were included in the analysis, as follows:

- **standard embeddings (id:STD)** - uses the standard randomly initialised embedding layer in the FastPitch architecture, jointly learned in the TTS training process;
- **TitaNet embeddings (id:TNET)** - the embedding layer is initialised with the mean embedding for each speaker as extracted by the NeMo TitaNet[3] architecture.
- **TitaNet frozen (id:TNET-FROZ)** - same as the previous, but the embedding layer is frozen during TTS training.
- **random frozen embeddings (id:RAND)** - the embedding layer is randomly initialised and frozen during training.
- **individual embeddings frozen (id:INDIV)** - uses the individual TitaNet-derived embeddings for each utterance in the dataset, instead of the mean embedding for each speaker;

The choice for the TitaNet architecture is based on the results of [17] where multiple speaker embedding models were jointly analysed in an effort to determine the amount of residual information present within them. The TitaNet-derived embeddings showed some of the best performances. No finetuning of the pretrained model[4] was performed such that future speaker identities used in the TTS would not require to retrain the embedding extractor. FastPitch uses 384-dimensional speaker embeddings, while TitaNet outputs a 192-dimensional representation. To feed these representations into FastPitch, we replicated them across the 0th axis.

The use of individual embeddings for each sample, instead of the average embedding for each speaker, is derived from the fact that the embeddings showed high correlations with out of distribution samples in the dataset. Therefore, we wanted to enclose within the embedding layer as much variation as possible. This would presumably enable the core network to learn a better abstraction of an eigen voice-like identity. At inference time, we randomly select one of the embeddings from the corresponding speaker and use it to condition the output.

All the above sets of embeddings were used to finetune the pretrained SWARA-based model for an additional 800k iterations. For the standard embeddings procedure, we also used the MARA pretrained model (id:MARA). This was to check if the pretrained model's training set has any effects over the output TTS quality. Same as before, the multi-speaker model was finetuned for an additional 800k iterations over the baseline, with a batch size of 8.

# 4. Results

## 4.1. Speaker identity preservation

A first analysis looked into how the speaker identity is maintained across all TTS systems. The cosine similarity between the TitaNet-derived embeddings of the synthesised samples and their respective vocoded counterparts was computed. We did not use natural samples, as the HiFi-GAN vocoder also introduces additional shifts in the SV-derived embeddings, and therefore wanted to alleviate the vocoder's influence over the results.

Results are shown in Figure 1 for each system. It can be

[3]https://huggingface.co/nvidia/
speakerverification_en_titanet_large
[4]https://catalog.ngc.nvidia.com/orgs/nvidia/
teams/nemo/models/titanet_large

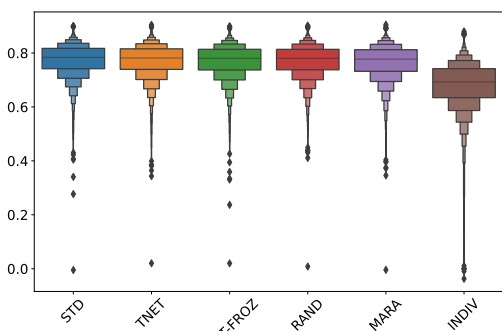

Figure 1: *Cosine similarity measure distribution for the multi-speaker synthesis systems. The similarity was measured based on the TitaNet-derived embeddings for each synthesised sample in the test set against its corresponding vocoded sample .*

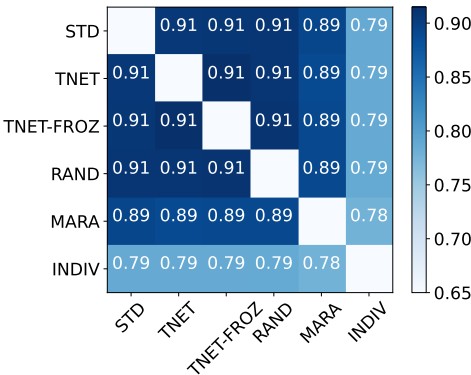

Figure 2: *Inter-system cosine similarity matrix for pairs of utterances from the different TTS systems. The similarity was measured using the TitaNet-derived embeddings for each synthesised sample.*

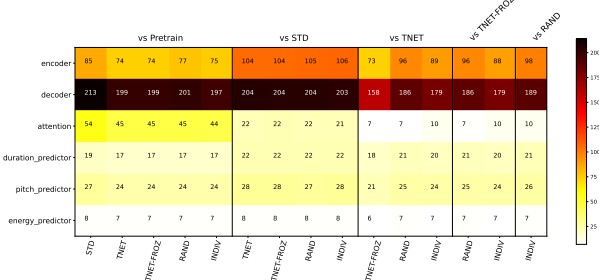

Figure 3: *Module weight differences between the multi-speaker and the pretrained model, as well as the differences between the multi-speaker models. The absolute values are normalised with respect to the number of parameters within the respective module.*

noticed that most systems exhibit similar results, and that the average cosine similarity is around 0.77. As reference, the intra-speaker similarity across pairs of natural samples for the TitaNet architecture is 0.75. One exception is the `INDIV` system, where the mean cosine similarity is 0.68, with more outliers. This was to be expected as the training of this system relied on individual embeddings for each sample, and this adds more complexity to the abstraction process within the network.

One speech sample consistently yielded very low similarity scores. At closer inspection, this sample was erroneous and contained just silence. The other outliers across the first five systems were short utterances (at most 4 words). Similar results were obtained after just 100 epochs of finetuning. The average cosine similarity was 0.72, with 0.68 for the `INDIV` system.

### 4.2. Inter-system correlation

To determine how the different embedding choices affect the quality of the output speech, we computed the inter-system similarity measures. Each sample from the test set was compared against the same sample from each of the other systems. The similarity matrix is plotted in Figure 2. It can be noticed that, with the exception of the `INDIV` system, all similarities are around 0.9. This was a surprising result, especially for the random embeddings choice where the network is conditioned on random vectors assigned to each speaker identity and frozen during training. This presumably shifts all speaker learning to the core network, with no significant information learning being transferred to the embedding layer.

In conjunction with the results of the previous subsection, this is an indication that, irrespective of the choice of embeddings and embedding training setup, the synthesised output is very similar across the systems, and has a high correlation to the target identity. It also means that, even if the embedding layer is not at all indicative of a speaker identity representation, the other modules of the network are able to accommodate this information. To see how this information is distributed within the network, we measure the weight changes across the major modules of the FastPitch architecture with respect to the pretrained models, and also between the different multi-speaker models. Figure 3 shows a visual representation of the absolute level of network weight shifts that occur in the: encoder, decoder, attention, duration predictor, pitch predictor and energy predictor modules. The values are normalised to the number of parame-

ters within each module. We leave out the `MARA` systems as it is based on a different pretrained model and a direct comparison would not be necessarily informative. The maximum changes pertain to the decoder weights, followed by the encoder and the attention. It is interesting to notice that all models are more or less equidistant to each other, yet generate very similar outputs. We can assume that the models landed in distinct local minima of the objective function, and that the speaker embedding choice might have, to some extent, led to this behaviour.

### 4.3. Zero conditioning

One desirable side-effect of training truly speaker identity disentangled TTS models is that the core model (i.e. zero conditioned) may learn an abstract representation of an eigen-like speaker derived from the training speaker data. The speaker conditioning in this case would only shift this representation towards the desired timbre and prosodic characteristics. This would also mean that the zero conditioned system should yield the same speaker identity for any input text.

We looked into this aspect of our trained models and generated the same set of samples as in the previous section, but used a zero input vector for the speaker embedding layer. We then plotted the SV-derived embeddings of the synthesised samples using t-SNE. The results are shown in Figure 4. The topline re-

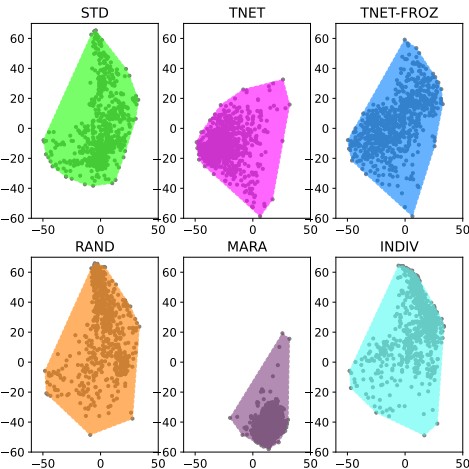

Figure 4: *t-SNE plots of the speaker embeddings across the TTS systems for the zero conditioning setup. The point distributions are overlapped with the surface polygons for better visualisation.*

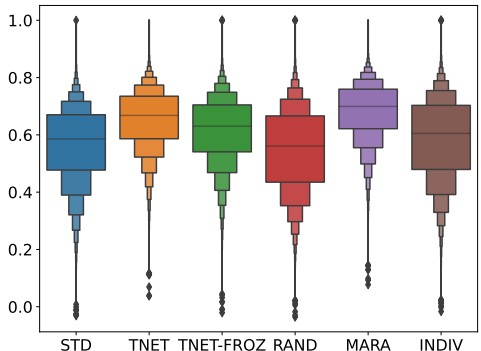

Figure 5: *Intra-system cosine similarity in the zero conditioning analysis. The similarity was measured using the TitaNet-derived embeddings for each synthesised sample.*

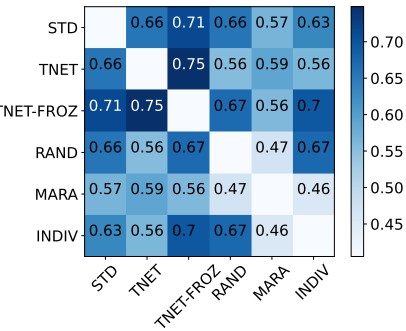

Figure 6: *Cosine similarity of pairs of corresponding samples across the TTS systems. The similarity was measured using the TitaNet-derived embeddings for each synthesised sample.*

sults would have exhibited a uniform cluster of points with limited standard deviation. This is only partially true for the `MARA` system, in which the pretrained model is a single speaker system. The `TNET` system also appears, to some extent, to create similar, more correlated sounding identities across the test samples. To estimate this effect, we also compute the cosine similarity measures between pairs of generated samples from the same system, and plot them in Figure 5. We see a much wider distribution of the similarity scores for all systems, with only marginal improvements for the `TNET` and `MARA` systems. By listening to the synthesised samples, we noticed that the output voice has a female-like identity with some minor shifts towards some of the speakers in the dataset. No definite identification of one of the speakers was possible–which is in line with the expected results.

However, we again need to take into consideration the fact that the SV network does not appear to truly disentangle the speaker representations. Therefore, we also analysed if there are any correlations between the samples across the systems in terms of their linguistic content. The results are shown in Figure 6. It appears that, given the same training data, and only changing the speaker embedding, the core network learns a rather similar representation of speech in the zero-conditioning scenario, with similarity measures above 0.6 for most pairs of systems.

The findings suggest that utilising a speaker conditioning layer in the network does not guarantee complete disentanglement of identity in the core architecture. Additionally, a simple conditioning on a speaker representation may not ensure a consistent behaviour of the synthesised speaker identity–which was also observed in the FastPitch architecture.

### 4.4. Layer freezing

In a separate analysis, we wanted to explore if different modules of the architecture can be directed towards speaker agnosticism.

We, therefore, froze all modules of the network, and fine-tuned only the embedding layer. In all our embedding layer initialisation scenarios (i.e. standard, transferred, random), the layer was unable to adapt to the variability of speaker infor-

mation required to generate the desired speaker identities. The synthesised samples had random identities with very little correlation to the target speaker. In a separate setup, we removed the conditioning from the text encoder, and only used the speaker embedding to condition the pitch, energy, and duration predictors. The model was trained using the standard embedding layer setup for 800k iterations. Again, the output did not follow the target speaker identity. A t-SNE plot of the embeddings extracted from the vocoded and generated samples is shown in Figure 7. We can see that the vocoded representations are clustered in terms of speaker identity, while there is limited to no clustering for the synthesised samples. Some speakers, such as 'bas', 'bea' and 'eme' partially show a clustering behaviour, with 'bas' being prominently disjointed from the large cloud of synthesised embeddings. By also listening to the generated samples, this behaviour is clearer, and different prompts conditioned on the same speaker generate seemingly random speech identities in this training scenario. This indicates that the disentangling process requires more computational bandwidth within the network – hence the speaker leakage within the encoder and decoder modules.

## 5. Conclusions

This paper introduced a set of analyses regarding the use of different types of speaker embeddings within a non-autoregressive TTS system. The analyses were founded on the idea that many

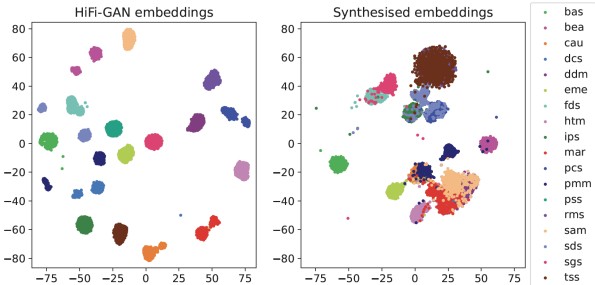

Figure 7: *t-SNE plots of the TitaNet embeddings for the vocoded samples (on the left), and the synthesised samples when conditioning only the variance adaptors in the network (on the right). Different colours pertain to the different speakers.*

novel research methods show that using different sets of speaker representations yield better results, especially in the zero- or few-shot learning strategies.

Six different embedding sets were used to condition a factorised neural architecture which was pre-trained on either a large set of speech data from multiple speakers, or on a single speaker dataset. The choice of embedding sets looked into standard trainable embedding layers and frozen embedding layers, as well as random or speaker verification-derived representations. A high quality dataset of parallel samples from multiple speakers was employed, thus limiting the effects of linguistic content and speaker representation in the training data within the generation of each voice identity.

Our results showed that the choice for embedding set does not affect the learning process in any way, and irrespective of this choice, the network is able to accommodate the speaker conditioning and yield high quality results. A side output of our study is related to the eigen-like identity learned by the core TTS architecture, that are the zero conditioned outputs in different evaluation scenarios. This analysis showed that speaker leakage is in the core modules, irrespective of the choice of embeddings, and that the task of speaker disentanglement requires an enhanced representation complexity compared to those used in recent literature.

As future work, we would like to investigate better means of restricting the speaker leakage within different modules of the network, and to also analyse if the zero-conditioned output can be exploited for more efficient voice cloning applications.

**Acknowledgement.** This project has received funding from the Horizon 2020 research and innovation programme under the Marie Skłodowska-Curie grant agreement No 859588.

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
