# OpenReview forum: "An analysis on the effects of speaker embedding choice in non auto-regressive TTS"
_Interspeech.org/2023/Workshop/SSW — SSW12_

### Official Review · Reviewer_QtJU · 2023-05-19
**An interesting paper analysing the use of different speaker identity embeddings**

**Rating:** 7
**Confidence:** 4

**Review:**

This paper investigates the implications of using different speakerID representations in TTS. Which is interesting in that I don’t think anyone has really done this beyond reporting the results of synthesis with and without using an externally trained SV based speakerID model.
Their selection of system configurations seems reasonable, and the TitaNet a suitable choice for an externally trained system.
I guess the results are a little surprising in that the representation doesn’t appear to make a great difference. The zero conditioning experiment is interesting, and would like to hear the samples here.
One thing I feel is missing from the paper is a subjective listening experiment where raters are given a pair of stimuli and asked if they are the same speaker. But given that the underlying representation all appear to perform similarly I think this is less important than it would be if there were differences.
A few things the paper doesn’t say and could.
How good is their underlying TTS system. What kind of MOS does it achieve?
The paperalso  doesn’t really address language as a factor, the experiments are carried out on Romanian data, would the same hold true for other languages? Also is there Romanian data in the pre-trained TitaNet model?
Quality: This work has been carried out to a good technical standard and appears to be correct in its findings.
Clarity: The paper is easy to understand, and should be reproducible.
Originality and significance of this work: This type of analysis is original and insightful for anyone doing multi-speaker TTS.

---

> ### Author Response · Authors · 2023-06-26
>
> Thank you for taking the time to review our paper. We made the following changes and answered some of your questions as described below:
>
> How good is their underlying TTS system. What kind of MOS does it achieve?
> - In our informal listening tests, all speakers showed on average a 4.1 MOS score. However, given the  large number of speakers and embedding sets it would have been unfeasible to perform a large listening test for this particular study.
>
> The paper also doesn’t really address language as a factor, the experiments are carried out on Romanian data, would the same hold true for other languages?
> - We chose the Romanian dataset mainly to its parallel characteristic, meaning that all speakers read the same set of prompts. This would factor out the linguistic component from the behaviour of the resulting network. We are definitely interested in performing a similar analysis on English and other languages data, but to the best of our knowledge, no other large multi-speaker parallel dataset is available.
>
> Also is there Romanian data in the pre-trained TitaNet model?
> - To the best of our knowledge and the reports from the original TitaNet paper, there was only English data used in the training. It is true that this may skew our results to some extent, but our main idea was to not finetune any model that we use such that future applications of that particular model would require finetuning data and computation.

---

### Official Review · Reviewer_qe7W · 2023-06-05
**A very thorough analysis of speaker embeddings/disentanglement in TTS (FastPitch)**

**Rating:** 8
**Confidence:** 4

**Review:**

Key Strengths
* This paper provides a very thorough analysis of the use of speaker embeddings with respect to FastPitch (i.e. non-autoregressive TTS).
* The experiments are very well motivated and generally well explained.  The visualisations are generally clear and helpful.
* The range of speaker embedding types/setups is comprehensive and well thought through.
* The investigation in leakage of speaker identity is very interesting and increasingly important as we move to more multispeaker models.  The lack of change observed with the different embedding types (including frozen random embeddings!), and subsequent capture of speaker features into the decoder (and other parts of the model) is a very important contribution.
* This is an excellent paper.  I would argue for it being included in the workshop.  I am very happy to see a paper that goes for depth in probing what is actually happening in neural TTS.

Main Weaknesses
* It's a little unclear at some point exactly what vectors are being compared, e.g. in 4.1-2 comparing speech samples, what exactly is being compared in the cosine similarity measurement? Later measurements are based on speaker verification networks.
* It's not clear how restricted these results are to FastPitch (but I don't think this should be considered barrier to the paper being accepted - it is fine to start with an in-depth analysis of one model).

Novelty/Originality
* The paper is highly relevant to the SSW audience and should be of interest to anyone interested in multispeaker TTS.  The work builds on and significantly extends previous work (most notably, Stan (2022) [18]).

Technical Correctness
* The paper is technically and scientifically solid.  The details are sufficient for reproducibility and provide a good roadmap for anyone else who would want to do similar investigations on other architectures and/or languages.

Suggestions for improvement
* Please check the inputs to similarity measures are clear.

Quality of References
* references are good.

Clarity of Presentation
The writing is generally clear and the argumentation makes sense.

---

> ### Author Response · Authors · 2023-06-26
>
> Thank you for taking the time to review our paper. We made the following changes and answered some of your questions as described below:
>
> * It's a little unclear at some point exactly what vectors are being compared, e.g. in 4.1-2 comparing speech samples, what exactly is being compared in the cosine similarity measurement? Later measurements are based on speaker verification networks.
> - More details have been added to the figure captions
>
>
> * It's not clear how restricted these results are to FastPitch (but I don't think this should be considered barrier to the paper being accepted - it is fine to start with an in-depth analysis of one model).
> - It is true that to a large extent, the DNN architecture may influence the behaviour/presence of the speaker leakage. However, we considered that FastPitch and its versions are quite widely adopted in the TTS community. Also, its non-autoregressive and disentangled characteristics make it appealing in studying the effects of various factors over the TTS output. We do plan to evaluate other deep architectures and examine if they exhibit similar results as future work.

---

### Decision · Program_Chairs · 2023-06-14

**Decision:**

Accept

**Comment:**

SSW2003 received 45 papers. The acceptance rate is 82%. We are pleased to inform you that your paper has been accepted by the SSW2023 Program Committee. Please read the reviews carefully and submit your camera-ready paper by June 28th. Most reviewers performed a detailed review. Please answer to their questions and consider their comments. Note that camera-ready papers are credited with one extra page to allow authors to consider reviewers’ suggestions. So max 7 pages in total including figures & refs.
The deadline for submitting the revised version (with full non-anonymized authors and refs!) is 28th June.